# The P-Site Loop of the Universally Conserved Bacterial Ribosomal Protein L5 Is Required for Maintaining Both Translation Rate and Fidelity

**DOI:** 10.3390/ijms241814285

**Published:** 2023-09-19

**Authors:** Mikhail G. Bubunenko, Alexey P. Korepanov

**Affiliations:** 1Basic Science Program, Leidos Biomedical Research Inc., Frederick National Laboratory for Cancer Research, National Cancer Institute, Frederick, MD 21702, USA; mikhail.bubunenko2@nih.gov; 2Institute of Protein Research, Russian Academy of Sciences, 142290 Pushchino, Russia

**Keywords:** *Escherichia coli*, ribosome, ribosomal protein uL5, P-site loop, translation fidelity

## Abstract

The bacterial ribosomal 5S rRNA-binding protein L5 is universally conserved (uL5). It contains the so-called P-site loop (PSL), which contacts the P-site tRNA in the ribosome. Certain PSL mutations in yeast are lethal, suggesting that the loop plays an important role in translation. In this work, for the first time, a viable *Escherichia coli* strain was obtained with the deletion of the major part of the PSL (residues 73–80) of the uL5 protein. The deletion conferred cold sensitivity and drastically reduced the growth rate and overall protein synthesizing capacity of the mutant. Translation rate is decreased in mutant cells as compared to the control. At the same time, the deletion causes increased levels of −1 frameshifting and readthrough of all three stop codons. In general, the results show that the PSL of the uL5 is required for maintaining both the accuracy and rate of protein synthesis in vivo.

## 1. Introduction

Bacterial ribosomal protein (r-protein) uL5 (here and further in the text, ribosomal proteins are named according to the guidelines proposed in [1]) is a conserved 5S rRNA-binding protein that has counterparts in all domains of life, which suggests that it plays a pivotal role in the protein synthesis. The uL5 protein is essential for survival in *Escherichia coli* [2] and is crucial for the in vivo assembly of the entire central protuberance of the large ribosomal subunit (LSU) including 5S rRNA and a number of r-proteins [3]. In the ribosome, uL5 forms numerous contacts with 5S rRNA, 23S rRNA, and r-protein uS13 (thus participating in the formation of the only protein–protein intersubunit bridge B1b) [4] and tRNA in the ribosomal P-site [5] (Figure 1). The latter contact is formed by the residues of the positively charged β2–β3 loop (the so-called P-site loop, PSL), which is structurally conserved, despite relatively low conservation of the whole protein sequence. This fact suggests that the PSL may perform an important conserved function during protein synthesis. It was demonstrated earlier that mutations introduced into the PSL of the yeast protein uL5, which is homologous to the bacterial uL5, affected translation fidelity, and the deletion of this loop is lethal. At the same time, to our knowledge, the role of the bacterial PSL in translation has not been dissected in detail. The present study is focused on constructing an *E. coli* strain bearing the mutant *rplE* chromosomal allele encoding uL5 protein with deleted amino acid residues 73–80 situated at the tip of the PSL (uL5ΔPSL). These residues contact the P-site tRNA and helix 84 of the 23S rRNA in the crystal structures of ribosomes. The results obtained suggest that while the constructed mutant is viable, the deletion causes slow growth and cold sensitivity. Moreover, the deletion slows down protein synthesis and reduces translation fidelity in vivo.

## 2. Results

### 2.1. Deletion of Residues 73–80 of uL5 Reduces Growth Rate and Causes Cold Sensitivity

To assess the effect of the deletion of residues 73–80 (the P-site loop, PSL) in the uL5 in vivo, two strains were constructed using highly efficient homologous recombination approach called “recombineering” (see Materials and Methods). The control strain, MS129, carries the wild type uL5 open reading frame (ORF) followed by chloramphenicol acetyltransferase ORF (*rplE1::cat*), whereas the isogenic mutant strain, MS129a, carries the *rplE* allele encoding the uL5 with the deletion of the residues 73–80 (*rplEΔ73–80::cat*). The generation time of the control strain MS129 (*rplE1::cat*) in the LB medium at 37 °C (21 ± 1 min) was virtually indistinguishable from that of the parental wild type MG1655 (20 ± 1). This suggests that the insertion of the *cat* ORF downstream of the *rplE* within the large *spc* operon did not affect the expression of the operon. The MS129a (*rplEΔ73–80::cat*) strain possessing the uL5 protein with the deletion of amino acid residues 73–80 (uL5ΔPSL) duplicated every 52 ± 4 min in the same conditions. Thus, under optimal growth conditions, the deletion of residues 73–80 reduces the growth rate by the factor of 2.5. 

The growth of the ΔPSL mutant on LB agar plates was also decreased at both 37 °C and 42 °C compared to the control. The mutant failed to form colonies at 23 °C within at least two weeks (Appendix A). Therefore, the deletion mutant possesses a clear cold sensitive phenotype.

### 2.2. Ribosomal Profiles and Protein Composition of Ribosomes Are Virtually the Same in the ΔPSL Mutant and the Control Strain

To assess the distribution of 70S ribosomes and free ribosomal subunits in mutant cells, cell lysates were fractionated by ultracentrifugation through a sucrose concentration gradient (Figure 2). The ribosomes of the mutant strain were predominantly in the 70S form, with only a marginal fraction of free subunits, which is similar to the control. 

No other major peaks were detected in the deletion mutant lysates, indicating that the large subunit (LSU) assembly intermediates are not accumulated there. The PSL deletion did not affect the association of mutant LSU with the small ribosomal subunit (SSU). Thus, LSU assembly in the mutant cells appears normal. At the same time, it is worth mentioning that the free subunits are slightly, but reproducibly, overrepresented in the mutant as compared to the control.

To further prove that LSU assembly is not affected in the ΔPSL cells, the protein composition of washed ribosomal fractions from the control strain MS129 and the mutant MS129a was assessed using 2D electrophoresis (Figure 3). The protein composition of ribosomes of the ΔPSL mutant was equal to that of the control. The only difference observed was the shift in the position of the spot corresponding to the mutant uL5 (uL5ΔPSL). This occurs because uL5ΔPSL lacks two positively charged amino acid residues, while the electrophoresis system applied is quite sensitive to protein charge. 

### 2.3. The Rate of Protein Synthesis Is Reduced in the ΔPSL Mutant Cells

The IPTG-induced synthesis of the β-galactosidase (LacZ) is a classical way to characterize protein synthesis in cells. Protein-synthesizing capacity, i.e., the total ability of cells to produce proteins, is commonly estimated by measuring the accumulation of LacZ during the linear phase of synthesis. The slopes of the LacZ accumulation curves at 37 °C (Figure 4) indicate that mutant strain synthesized proteins at least twofold less efficiently than the control strain, consistent with the growth rate reduction in the mutant strain. 

Using the same approach but looking at the very early stages (within the first several minutes after the induction of LacZ synthesis), one can calculate the time period required for the synthesis of the (first) LacZ molecule by plotting the square root of LacZ activity vs. time. The results of a typical experiment are shown in Figure 5: the synthesis of the LacZ monomer takes 79 ± 2 s (approximately 13 amino acid residues per second) in the control strain and 89 ± 2 s (~11.5 aa/s) in the mutant. 

To see the possible impact of low temperature on the rate of protein synthesis in the deletion mutant, cell cultures grown at 37 °C were shifted to 25 °C, allowed to equilibrate for 5 min, and then, the LacZ synthesis was induced. Figure 6 shows that the time period needed for the synthesis of a LacZ chain at this temperature is 193 ± 4 s (about 5.3 aa/s) in the control strain and 268 ± 6 s (3.8 aa/s) in the deletion mutant (here and further in the text, mean values and standard deviations are given); that is, at 25 °C, the mutant strain synthesizes proteins as much as 35% slower than the control. Since mutant ribosomes elongated only about 12% slower than the control ones at 37 °C, it is reasonable to conclude that the cold sensitivity of the mutant strain caused by the PSL deletion is primarily due to the translation elongation defects, rather than due to the defects in the ribosome biogenesis: after a 5 min shift to 25 °C prior to the induction of LacZ synthesis, new ribosomes were not produced to an extent sufficient to affect the results of this experiment.

### 2.4. Translation Fidelity Is Reduced in the ΔPSL Mutant

The decrease in the protein synthesis rate in the ΔPSL mutant is too small to be solely responsible for 2.5-fold reduced growth rate. Another potential problem could be compromised translation fidelity. To estimate translation fidelity in the ΔPSL mutant, six MG1655-based *E. coli* strains carrying reporter *lacZ* fusions in the chromosomal *lac* locus were constructed. These fusions comprise mutant *lacZ* ORFs containing +1, −1 frameshifts or premature stop codons: UAA, UAG, and UGA, based on the sequences published in [7]. A corresponding “wild type” *lacZ* fusion was constructed, which was used for the normalization of the results. Then, the *rplE1::cat* (wild type uL5) and *rplEΔ73–80::cat* (uL5ΔPSL) alleles were transferred into these reporter strains. The LacZ activity is proportional to the frequency of frameshifting and readthrough of stop codons present in the corresponding *lacZ* variants. LacZ activity was measured in exponentially growing cultures. To compare the activities of each dissected fusion, the results were represented as a ratio of activity of the reporter *lacZ* fusion variant to the activity of the control “wild type” *lacZ* fusion (Figure 7) in the corresponding genetic background (control *rplE1::cat* or mutant *rplEΔ73–80::cat*). 

The experimental data indicate that probability of −1 frameshifting in the mutant is about twice as high as in the control, whereas levels of +1 frameshifting is approximately the same in both strains. The levels of readthrough of all the three premature stop codons were also dramatically increased in the ΔPSL mutant. Depending on the stop codon, the ΔPSL mutant strain demonstrated a 3- to 7-fold increase in the levels of stop codon readthrough (for more details see Appendix A). 

## 3. Discussion

Ribosomal protein uL5 is present in all domains of life. Though, its sequence is not highly conserved (Appendix A), especially between Bacteria and Eukarya, the three-dimensional structures of uL5 proteins from different species are quite similar. In particular, the structure of the β2–β3 loop (residues 70–83 in *E. coli*), the so-called “P-site loop” (PSL) is conserved in available structures of ribosomes (Appendix A). The PSL carries a net positive charge, which probably facilitates its interaction with the negatively charged sugar-phosphate backbone of the P-site tRNA and helix 84 of the 23S rRNA visible in numerous crystal structures of ribosomes (Appendix A). The conservation of the PSL structure suggests that the loop plays an important conserved role in protein synthesis. To the best of our knowledge, the role of the bacterial PSL in translation has never been assessed in detail. Although, there are data indicating that the complete PSL deletion might cause lethality in *E. coli* cells [8]. According to the available data [9], in yeasts, the deletion of the entire PSL or deletion (as well as replacing to alanines) of residues 57–60, which correspond to residues 77–80 in the *E. coli* uL5, including invariant F77 and R80, are lethal. The authors suggested that the PSL and in particular these four residues are essential for the viability of the yeast cell. However, this is not the case in bacteria: a viable ΔPSL mutant with the deletion of the eight residues 73–80 constituting most of the PSL can be obtained at high recombination efficiency. This suggests that the bacterial PSL is not required for cell survival (see [10]).

Although the deletion of the PSL is not lethal in bacteria, this region of uL5 plays an important role in the bacterial cell physiology: the ΔPSL mutant strain grows approximately 2.5 times slower than the control strain in a liquid LB medium at a permissive temperature (37 °C). The ΔPSL strain demonstrates slow growth at both 37 °C and 42 °C and fails to form colonies at 23 °C (Appendix A), i.e., it is cold-sensitive. 

The uL5 protein is situated in the central protuberance of the large ribosomal subunit (LSU) facing the head of the small ribosomal subunit (SSU). It participates in the formation of the only protein–protein intersubunit bridge B1b by contacting SSU protein uS13 [4] (Appendix A). This bridge is important for ribosomal subunit association [11]. The analysis of available high-resolution structures of bacterial ribosomes (ribosomes from *T. thermophilus* are presently the most thoroughly studied ones) indicates that the PSL wedges itself between the elbow of the P-site tRNA and helix 84 of the 23S rRNA. Thus, the PSL mediates interaction between the P-site tRNA and H84. H84 is one of the mobile 23S rRNA helices, which directly contacts helix 38 of the 23S rRNA (the so-called A-site finger, ASF) and their movements are coordinated during the elongation cycle [12]. Therefore, the PSL participates in the formation of a physical link between tRNAs in the ribosomal A- and P-sites, which may potentially contribute to the coordinated relocation of tRNA molecules on the ribosome during translation in addition to similar role proposed earlier for the 5S rRNA [13].

During in vivo assembly of the LSU, uL5 is absolutely required for the incorporation of the 5S rRNA and a number of LSU r-proteins located in the central protuberance [3]. Thus, after introducing the deletion into the uL5 protein, we first ensured that there were no structural changes in ribosomes, which could contribute to the ΔPSL mutant cell phenotype or to the observed functional defects in the translation machinery of the ΔPSL mutant strain. Our data indicate that mutant LSU are able to associate with SSU to form 70S ribosomes (Figure 2), and we did not observe the accumulation of any LSU assembly intermediates. We repeatedly observed that the free subunit content in the lysates of the ΔPSL mutant is slightly higher as compared to the control strain. Although, such difference is negligible to be accounted for the observed physiological and functional defects in the mutant, it still may indicate that some small fraction of the ΔPSL LSU is somewhat compromised for the association with the SSU, at least, under the experimental conditions used to isolate and analyze ribosomes (low temperature, etc.). These results suggest that the PSL deletion does not significantly affect the assembly of the bacterial LSU, which was further confirmed by checking the protein composition of mutant ribosomes.

Therefore, the effects observed in ΔPSL mutant cells are likely to be due to lack of the contacts formed by the eight deleted PSL residues in the translating ribosome. 

The total protein synthesizing capacity of the mutant cells was about the half of that of the control (Figure 4), which agrees with the results of the cell growth experiments. Therefore, the slow growth of the ΔPSL mutant can be explained by the defects in protein synthesis, that is, by affecting ribosome functions. At the same time, the rate of protein synthesis was only marginally (about 12%), but reproducibly, decreased in mutant cells at 37 °C (Figure 5). This difference was dramatically increased at 25 °C (Figure 6). Thus, the translational defect observed in the ΔPSL mutant strain even at 37 °C becomes more significant at lower temperature, which agrees with the cold sensitivity of the mutant. The 12% deceleration of the protein synthesis cannot explain per se the 2.5-fold growth defect caused by the PSL deletion at 37 °C.

To further characterize protein synthesis in the mutant cells, the levels of +1 and −1 frameshifting were measured along with the levels of stop codon readthrough using newly constructed chromosomal reporter *lac* fusions based on the plasmid-borne ones described in [7]. Using this system eliminates the problem of the fusion allele copy number, as it is only present in the chromosomal *lac* locus. Another advantage is that this system does not require using antibiotics to preserve the fusions in cells.

The level of +1 frameshifting was essentially the same in the ΔPSL mutant and in the control, whereas the level of −1 frameshifting in the ΔPSL mutant was at least twice that of the control (Figure 7). Both +1 and −1 frameshifts are kinetically-driven events, but the first requires a vacant ribosomal A-site, whereas the latter event occurs only when the A-site is occupied [14,15]. According to the current “peptidyl-tRNA slippage” model, frameshifting occurs in the P-site; it requires the destabilization of codon–anticodon pairing and translational pausing, which provides sufficient time for anticodon relocation (reviewed in [15,16]). Unlike the A-site tRNA, the position of the P-site tRNA is stabilized through extensive contacts with surrounding P-site regions, including 16S rRNA, 23S rRNA, and r-proteins, which is important for peptide transfer and for maintaining the reading frame [16]. Indeed, a number of tRNA mutations, which destabilize the elbow region, i.e., the tRNA region that contacts the PSL (Appendix A), modulate codon–anticodon slippage, and some of them were shown to induce −1 frameshifting [17]. This effect is similar to what was observed in this work for the PSL deletion. Thus, the contact between the elbow region of peptidyl tRNA and uL5 (and H84 of the 23S rRNA) is important for the stabilization of the P-site tRNA position and for maintaining the reading frame. As it was mentioned above, frameshifting also requires translational pausing. Indeed, in our experiments, translation rate in the ΔPSL strain was 12% reduced compared to the control at 37 °C. It seems reasonable to propose that the destabilization of the P-site tRNA caused by the PSL truncation affects the positioning of the acceptor stem and thus interferes with peptide transfer. We speculate that this can explain the slower translation rate in the ΔPSL cells, although finding the exact translation step affected by PSL deletion is a matter of further detailed studies.

The 3- to 7-fold increase in the levels of readthrough of all three premature stop codons (Figure 7) correlates with a decreased translation accuracy observed in the case of certain PSL mutations in yeast [9]. Thus, the data suggest that compromised translation fidelity is a major reason for the slow growth of the ΔPSL mutant and less efficient synthesis of active LacZ. Although mRNA decoding occurs within the decoding center (i.e., in the A-site of the SSU) and is not obviously related to the tRNA binding in the P-site of the LSU, there is long-known interplay (inverse correlation) between the strength of the aminoacyl-tRNA (aa-tRNA) binding to the A-site and the peptidyl-tRNA binding to the P-site [18]. The error-prone (Ram) ribosomal mutations destabilize peptidyl tRNA at the P-site. On the contrary, the P-site-tRNA binding is stabilized in hyper-accurate (SmD) ribosomes. The opposite effect is observed for the A-site aa-tRNA affinity [19]. The stabilization of aa-tRNA binding by non-codon specific interactions in the A-site facilitates the acceptance of near(non)-cognate aa-tRNAs at the proofreading step leading to translation errors. Thus, the increased nonsense-suppression in the ΔPSL mutant could be explained by higher affinity of aa-tRNA to the A-site induced by the destabilization of peptidyl tRNA at the P-site. The increased rate of -1 frameshifting in the ΔPSL mutant also speaks in favor of higher affinity of aa-tRNA to the A-site in the mutant compared to the control, as it requires higher occupancy of the A-site. 

It is commonly recognized that the termination of protein synthesis on the three stop codons is a result of the competition between the readthrough via erroneous recognition of a stop codon by a non-cognate aa-tRNA and peptidyl-tRNA hydrolysis by the protein translation termination factors (RF1 and RF2 in bacteria). In theory, the observed increased readthrough of the three stop codons in the ΔPSL mutant could also result from the reduced affinity of the RFs for the ribosomal A-site and thus reduced translation termination efficiency. Apparently, there are data suggesting that some ribosomal RNA mutations in the ribosomal A-site could reduce translation termination efficiency without affecting the translation fidelity [20,21]. Although, we are not aware of such mutations within the P-site on the LSU, one cannot completely rule out the possibility that such mechanism also contributes to some extent to the stop codon readthrough in the ΔPSL mutant. Further detailed in vitro and in vivo experiments are required to establish that.

Summarizing the above, we speculate that the lack of contacts between H84 and the elbow region of the peptidyl tRNA at the ribosomal P-site (mediated by the PSL) in the ΔPSL mutant facilitates peptidyl tRNA swinging around the anticodon stem-loop (ASL)-CCA axis (Appendix A, right panel), thus destabilizing both ASL and CCA. The destabilization of the acceptor end interferes with the peptide bond formation resulting in the observed slowed translation and allowing more time for −1 frameshifting, which is provoked by the destabilization of ASL (i.e., codon–anticodon interaction). The destabilization of the peptidyl tRNA also induces increased A-site aa-tRNA affinity leading to decreased translation fidelity (nonsense codon suppression).

Interestingly, protein synthesis on the ΔPSL ribosomes features both low accuracy and somewhat reduced translation rate. This is quite unusual as, typically, the translation rate inversely correlates with the accuracy (reviewed in [22]). That is, increased translation rate correlates with decreased translation accuracy. An explanation of this phenomenon requires further detailed studies aimed at finding the specific translation step(s) affected by the PSL truncation. 

The results of the present studies suggest that the PSL plays an important conserved role in the translation in all kingdoms of life: both in yeast [9] and in the bacterium *E. coli*, the PSL is required for maintaining translation fidelity. 

## 4. Materials and Methods

Oligonucleotides (oligos), plasmids and bacterial strains obtained or used in this work are listed in Appendix A. Where appropriate, antibiotics were used at concentrations of 100 μg/mL (ampicillin) and 20 μg/mL (chloramphenicol) for selection purposes. To generate insertions and deletions in the *E. coli* chromosome, we used the “recombineering” technique [23]. This approach is based on a high efficiency recombination provided by bacteriophage λ Red functions. It allows replacing a target chromosomal sequence with a desired linear DNA construct carrying short (39–40 nucleotides) 5′-end and 3′-end extensions that are identical to sequences flanking the target. A standard generalized transduction by phage P1 [24] was utilized to exchange alleles between *E. coli* strains.

### 4.1. Introducing the Deletion in the Chromosomal rplE Gene

In *E. coli*, uL5 protein is encoded by the third cistron of the large *spc* operon. The 5′ portion of the corresponding *rplE* ORF encodes the regulatory element required for the feedback regulation of *rplE* and downstream genes [25]. First, the 3′ portion of *rplE* was first replaced with the open reading frame (ORF) of chloramphenicol transferase (*cat*) so that the first 34 codons of *rplE* were fused in frame to *cat*. To this end, the *cat* ORF was PCR-amplified using oligos rplE34cat-F1 and rplEcat-R1 and the pACYC184 plasmid [26] as a template. This *cat* PCR cassette was recombined into the NM300 strain, which carries the mini-λ-Tet prophage [27] and the complementing pNK12 plasmid (Appendix A) that provides an arabinose-induced synthesis of uL5. Recombinants were selected on LB agar plates supplemented with chloramphenicol and 0.2% arabinose at 32 °C. The resulting strain carrying the *rplE* deletion *ΔrplE34::cat*, *mini-λ-Tet* and pNK12 plasmid was referred to as MS128/pNK12. 

The deletion of the *rplE* portion encoding amino acid residues 73–80 was first introduced into the plasmid-borne gene via PCR mutagenesis using QuikChange^®^ Site-Directed Mutagenesis Kit (Stratagene, La Jolla, CA, USA) with a pair of oligos L5delta73-80F and L5delta73-80R and pNK12 as a template. The mutated *rplE* variant of the resulting plasmid (referred to as pNK14) and the wild type *rplE* (of pNK12) were PCR-amplified with oligos rplE_rec_F and rplE_rec_R. The DNA fragments obtained were recombined into the MS128/pNK12 selecting recombinants on LB agar plates supplemented with ampicillin at 37 °C. The selected MS128/pNK12 derivatives carrying either *rplE1::cat* (wild type uL5) or *rplEΔ73–80::cat* (deletion variant of uL5, uL5ΔPSL) were further purified on the arabinose-containing plates to avoid selecting second recombination events between the plasmid-borne wild-type *rplE* and chromosomal *rplEΔ73-80* allele. In view of potential instability of the MS128/pNK12 derivatives, they were not kept, and the *rplEΔ73–80::cat* allele was immediately transferred into MG1655 [28] by P1 transduction using *cat* selectable marker. The MG1655 transductants carrying either *rplE1::cat* or *rplEΔ73–80::cat* alleles were referred to as MS129 and MS129a, respectively.

### 4.2. Constructing Reporter Chromosomal Lac Fusions for Dissecting Translation Fidelity In Vivo

To provide reasonable expression levels of the *lacZ* reporter translation fusions carrying the premature stop codons or frameshifts, a system for high expression of chromosomal *lacZ* gene was constructed. A short DNA fragment made by PCR with mutually complementary oligos AK45 and AK46 was used as a “forward oligo” for the second round PCR with AK47 oligo and freshly made bacteriophage T4 lysate as a template. The resulting PCR cassette contained a “perfect” consensus −35 and −10 promoter boxes (based on the *trc* promoter), a part of the phage T4 gene 32 (positions −71 to +45, where +1 is A of the initiator AUG codon), which is known to stabilize mRNA [29] and short 5′- and 3′-extensions required for λ Red-mediated recombination. This PCR product was recombined into MS02 [30] selecting recombinants on sucrose-containing low-salt LB agar plates supplemented with ampicillin at 32 °C. The resulting strain containing the desired “wild-type” *lacI’::bla-Ptrc-g32-lacZ* construction in the chromosomal *lac* locus was referred to as MS45. It produced about 60,000 Miller units of LacZ, which severely inhibited cell growth. Thus, several spontaneous mutants were selected producing less LacZ and growing reasonably fast. One of these mutants (MS45a) that contained the insertion of G in the −35 promoter sequence (this Ptrc version was called Ptrc*) and expressed LacZ at about 20,000 Miller units was chosen for further fusion construction. The *lac* fusion region of this strain was PCR-amplified with the oligos AK07 and AK226. The DNA fragment obtained was used as a “forward oligo” for PCR amplification of cloned *lacZ* regions of plasmids pSG25 (WT *lacZ*), pSGlac7 (+1 frameshift), pSG12DP (−1 frameshift), pSG853 (premature UAA), pSG163 (premature UAG), and pSG3/4 (premature UGA) with oligo AK08b [7]. The six DNA fragments were recombined into MS02 as described above, giving rise to strains OK391a, OK391b, OK391c, OK391d, OK391e, and OK391f, respectively. The obtained *lacI’::bla-Ptrc*-g32-lacZ* variants (their sequences are provided in Appendix A) were transferred to MG1655 strain selecting transductants on agar plates supplemented with ampicillin and sodium citrate, thus generating strains OK392a, OK392b, OK392c, OK392d, OK392e, and OK392f, respectively.

### 4.3. Cell Growth and Experiments on Protein Synthesis and Translation Fidelity

To estimate cell growth rate, freshly prepared overnight cultures of the control MS129 and the mutant MS129a strains were 1:500 diluted in fresh LB medium and incubated at 37 °C with vigorous shaking. The optical density (OD) of the exponentially growing cultures was monitored at 600 nm in the OD_600_ interval 0.05 to 0.4. Generation times were calculated from the slope of the growth curve in a semi-log plot. The growth of the Δ73–80 mutant at different temperatures was estimated by growing the control MS129 strain along with the mutant on LB agar plates at 23 °C, 37 °C, and 42 °C. 

Protein synthesizing capacity of the cells was measured using the standard β-galactosidase assay as described in [24]. Overnight cultures were 1:500 diluted in minimal A medium supplemented with 0.5% glycerol and incubated until OD_600_ = 0.2. Then, IPTG was added to the final concentration of 1 mM, and aliquots were taken at indicated time points for determining β-galactosidase (LacZ) activity. To dissect translation elongation rate in the control and mutant strains, the time required for the synthesis of the LacZ molecule (1024 amino acid residues) was calculated by looking at LacZ accumulation during the parabolic synthesis phase [31], usually within the first three minutes. The square root of the LacZ activity after the deduction of basal activity (E(t)–E(0)) was then plotted versus the induction time. The x-intercept of the linear trendline indicates the LacZ synthesis time.

Total protein synthesizing capacity was estimated by the slope of LacZ accumulation curve at the linear synthesis phase. 

For the translation fidelity experiments, overnight cultures were 1:500 diluted in LB medium and incubated with shaking at 37 °C. Aliquots for measuring the LacZ levels were taken in duplicates at OD_600_ = 0.2 and OD_600_ = 0.4. 

### 4.4. Analysis of Ribosomal Profiles and Protein Composition of Ribosomes

The distribution of 70S ribosomes and nonassociated 50S and 30S subunits was dissected by ultracentrifugation of cell lysates in sucrose concentration gradients essentially as described in [3]. Ribosomal fraction was obtained via high-salt sucrose cushion wash as described in [32]. Ribosomal proteins were extracted with acetic acid according to [33] and analyzed by 2D electrophoresis in acidic–acidic system IV [34].

## Figures and Tables

**Figure 1 ijms-24-14285-f001:**
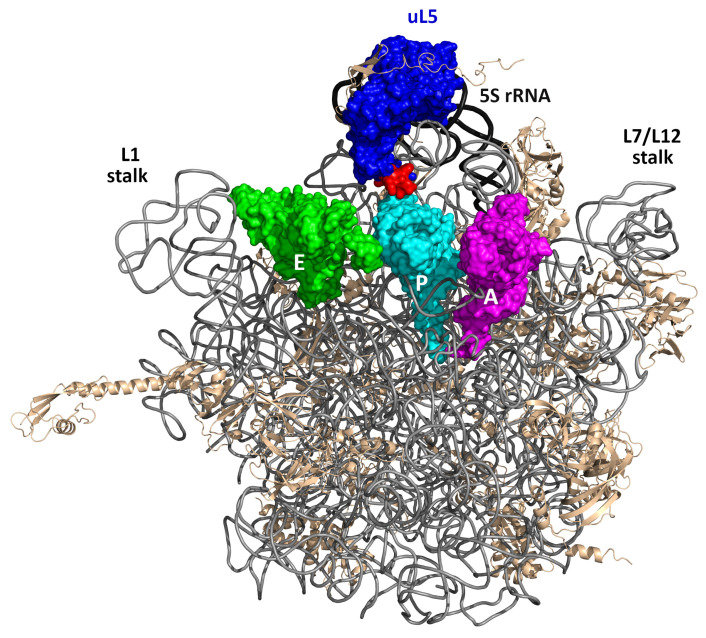
The bacterial 50S ribosomal subunit in the crown view with the three tRNAs in the E-, P-, and A-sites (green, cyan, and magenta, respectively). The deleted part of the P-site loop of uL5 (blue), contacting the P-site tRNA, is shown in red. The 23S rRNA, 5S rRNA, and r-proteins other than uL5 are shown in gray, black, and wheat, respectively. The figure was generated using the PyMOL Molecular Graphics System ver. 2.5; it is based on the crystal structure of the *Thermus thermophilus* 70S ribosome [6] (PDB entry 5DOY).

**Figure 2 ijms-24-14285-f002:**
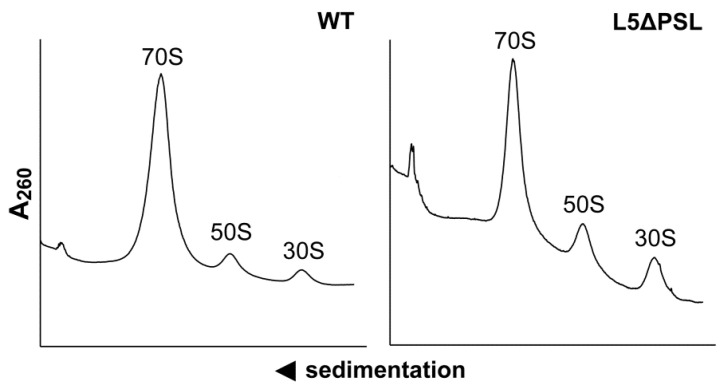
Distribution of 70S ribosomes and individual ribosomal subunits in lysates of the control strain MS129 (WT, left panel) and the mutant strain MS129a (uL5ΔPSL, right panel). Lysates were fractionated by ultracentrifugation in sucrose concentration gradients (5–20% *w*/*v*). Peaks corresponding to 70S ribosomes, large (50S), and small (30S) ribosomal subunits are indicated.

**Figure 3 ijms-24-14285-f003:**
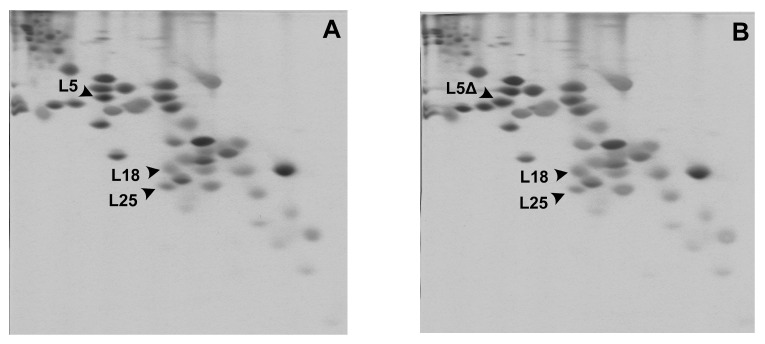
Protein composition of ribosomal fraction from the control strain MS129 (**A**) and the mutant strain MS129a (**B**). Spots of 5S rRNA-binding proteins uL5 (L5), its uL5ΔPSL variant (L5Δ), uL18 (L18), and bL25 (L25) are indicated with arrows.

**Figure 4 ijms-24-14285-f004:**
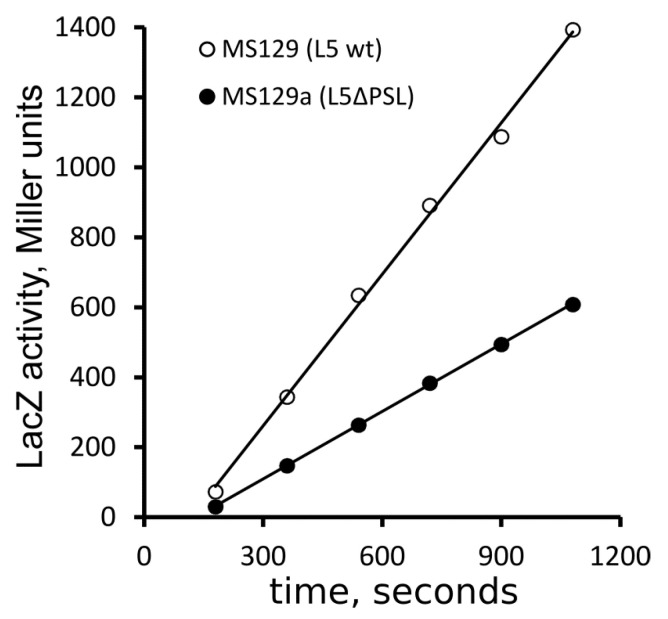
Accumulation of LacZ in the control MS129 and the mutant MS129a strains after induction with IPTG at 37 °C.

**Figure 5 ijms-24-14285-f005:**
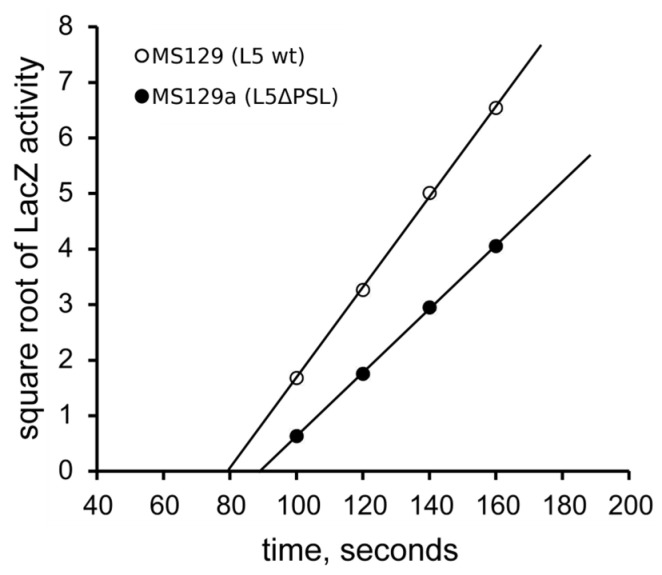
Rate of the protein synthesis in the ΔPSL mutant strain at 37 °C. Square root of LacZ activity (in Miller units) plotted vs. time after IPTG induction for the control strain MS129 (uL5 wt) and mutant strain MS129a (uL5ΔPSL). The x-intercept indicates time needed for the in vivo synthesis of the complete LacZ chain.

**Figure 6 ijms-24-14285-f006:**
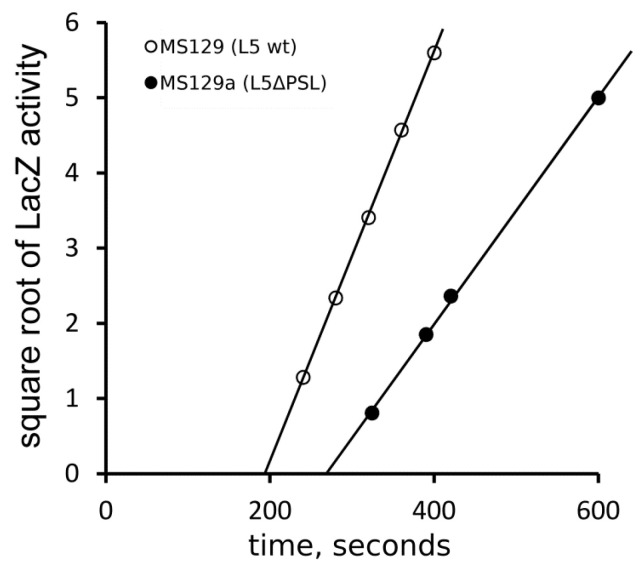
Rate of the protein synthesis in the ΔPSL mutant strain and the control strain at 25 °C. Other details are as in Figure 5.

**Figure 7 ijms-24-14285-f007:**
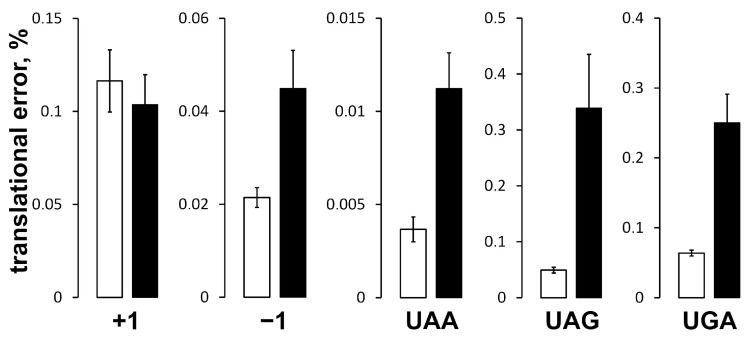
Translational errors in the control strain MS129 (open bars) and the uL5ΔPSL mutant MS129a (solid bars). Activity of the reference “wild type” *lacZ* fusion in the corresponding *rplE* variant strain was taken as 100%.

## Data Availability

The data supporting the findings of this manuscript are available from the corresponding author upon reasonable request.

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
