# Peer review of "The P-Site Loop of the Universally Conserved Bacterial Ribosomal Protein L5 Is Required for Maintaining Both Translation Rate and Fidelity"

_ijms, 2023, doi:10.3390/ijms241814285_

Round 1

Reviewer 1 Report

Bubunenko and Korepanov  constructed a P-loop deletion construct of uL5 in E.coli and analysed the impact of this deletion on protein synthesis and survival. They show that the deletion leads to a slower growth rate, a sensitivity to cold, and reduced protein synthesis. Further the mutant exhibits a higher degree of frameshifting events in comparison to the wild type. In general the study is well conducted and well presented. At some times further explanations (why, how) to the presented results would help the reader to follow. 

Some suggestions:

Line 48: A sentence detailing the technique used for the strain generation would help. control experiment. 

The Experiments shown in in Figure 3-5 need statistics (Mean/median and standard deviation), if no statistics are provided a statement why they are not necessary is needed.  

The article would greatly benefit from an introductory image showing the position of L5 within the ribosome, with a close-up view on the contact site with the P-site tRNA and the deleted part constructed for this study. 

Some explanations come very late in the discussion (e.g. why the 2D gel (Figure 1) was done. It would help to move these explanations further up into the result section. 

Author Response

Comments and Suggestions for Authors

Bubunenko and Korepanov  constructed a P-loop deletion construct of uL5 in E.coli and analysed the impact of this deletion on protein synthesis and survival. They show that the deletion leads to a slower growth rate, a sensitivity to cold, and reduced protein synthesis. Further the mutant exhibits a higher degree of frameshifting events in comparison to the wild type. In general the study is well conducted and well presented. At some times further explanations (why, how) to the presented results would help the reader to follow.

 Some suggestions:

Line 48: A sentence detailing the technique used for the strain generation would help. control experiment.

Authors’ answer:

Unfortunately, our version of the MS provided by IJMS for revision does not contain the line numbers (although we see that the line numbers are set and should be seen), but we have added two sentences to the section 2.1., explaining briefly the approach used to make strains and providing more details on the nature of the changes made. Hopefully, this is what we were suggested to do. Anyway, there is no need now to refer to the Materials and Methods section to understand those important things. Thank you.

Added text:

“To assess the effect of the deletion of residues 73-80 in the uL5 in vivo, two strains were constructed using highly efficient homologous recombination approach called “recombineering” (see Materials and Methods). The control strain, MS129, carries the wild type uL5 ORF followed by chloramphenicol acetyltransferase ORF (rplE1::cat), whereas the isogenic mutant strain, MS129a carries the rplE allele encoding the uL5 with the deletion of the residues 73-80 (rplEΔ73-80::cat).”

The Experiments shown in in Figure 3-5 need statistics (Mean/median and standard deviation), if no statistics are provided a statement why they are not necessary is needed. 

Authors’ answer:

For the Fig. 3, we provide just a typical experimental curve as for our purposes stating that the difference is never less than twofold. We believe that such an approximation is sufficient for our further consideration. As to the Figures 4 and 5, statistics is much more important, of course, and we provide the means and standard deviations in the text nearby. The text itself is quite laconic, so we dared not to repeat the numbers in the figures or the legends. We modified the main text, so that it’s now seen that we provide means and the standard deviations. It is clearer now, thank you.

Added text in the middle of the 3rd paragraph of the section 2.3:

“here and further in the text mean values and standard deviations are given”

The article would greatly benefit from an introductory image showing the position of L5 within the ribosome, with a close-up view on the contact site with the P-site tRNA and the deleted part constructed for this study.

Authors’ answer:

We agree, the Figure 1 and a corresponding legend were added at the end of the Introduction. The more detailed view of the contacts of uL5 may still be seen in the figure S4. It looks more informative, indeed.

Some explanations come very late in the discussion (e.g. why the 2D gel (Figure 1) was done. It would help to move these explanations further up into the result section.

Authors’ answer:

We carefully assessed one more time whether the reasons to apply the experimental assays chosen in this work are clearly explained and justified in the Results section of the manuscript, as suggested by the reviewer. We believe that the given information is sufficient, albeit concise, which makes the paper easier to read and understand. For instance, the specific example provided by the reviewer about the application of the 2D-electrophoresis is described as follows: To further prove that LSU assembly is not affected in the ΔPSL cells, the protein composition of washed ribosomal fractions from the control strain MS129 and the mutant MS129a was assessed using 2D electrophoresis (lanes 88-90 of the Results section). In the same concise but clear manner we presented the reasons behind the application of the sucrose gradient (e.g. lines 73-75) and all the functional tests used in the work.

Reviewer 2 Report

In this manuscript, authors show the effect of deletion of the P-site loop of ribosomal protein L5. Data clearly show that deletion of the P-site loop affects cell growth, cold sensitivity, translation rate and fidelity. 

Specific points: 

Although reduced affinity of class I peptide release factor (RF1 or RF2) for the ribosome can be a cause of readthrough, RF is not mentioned in the manuscript. 

Lines 12-13, lines 35-39

“PSL mutations in yeast are lethal” in Abstract is apparently inconsistent with “mutations introduced into the PSL of the yeast protein uL5, which is ---, affected translation fidelity” in Introduction.  

Lines 61-73, Figure 1

Authors claim that deletion of the P-site loop does not affect subunit assembly, as the ratio of the 70S peak area to 50S or 30S peak is similar. However, the ratio of the 70S to the subunit peaks in WT appears higher than in L5ΔPSL, especially in terms of peak area. This might not be ignored. 

Author Response

Comments and Suggestions for Authors

In this manuscript, authors show the effect of deletion of the P-site loop of ribosomal protein L5. Data clearly show that deletion of the P-site loop affects cell growth, cold sensitivity, translation rate and fidelity.

 Specific points:

 Although reduced affinity of class I peptide release factor (RF1 or RF2) for the ribosome can be a cause of readthrough, RF is not mentioned in the manuscript.

Authors’ answer:

We have added a 10th paragraph to the Discussion.

“It is commonly recognized that the termination of protein synthesis on the three stop codons is a result of the competition between the readthrough via erroneous recognition of a stop codon by a non-cognate aa-tRNA and peptidyl-tRNA hydrolysis by the protein translation termination factors (RF1 and RF2 in bacteria). In theory, the observed increased readthrough of the three stop codons in the ΔPSL mutant could also result from the reduced affinity of the RFs for the ribosomal A-site and thus reduced translation termination efficiency. Apparently, there are data suggesting that some ribosomal RNA mutations in the ribosomal A-site could reduce translation termination efficiency without affecting the translation fidelity [20, 21]. Although, we are not aware of such mutations within the P-site on the LSU, one cannot completely rule out the possibility that such mechanism also contributes to some extent to the stop codon readthrough in the ΔPSL mutant. Further detailed in vitro and in vivo experiments are required to establish that.”

Lines 12-13, lines 35-39

“PSL mutations in yeast are lethal” in Abstract is apparently inconsistent with “mutations introduced into the PSL of the yeast protein uL5, which is ---, affected translation fidelity” in Introduction. 

Authors’ answer:

It is misleading, indeed. We modified the sentence in the Abstract to point out that not all but at least some mutations in the yeast PSL are lethal.

This sentence in the Abstract now:

“Certain PSL mutations in yeast are lethal suggesting that the loop plays an important role in translation.”

 Lines 61-73, Figure 1

Authors claim that deletion of the P-site loop does not affect subunit assembly, as the ratio of the 70S peak area to 50S or 30S peak is similar. However, the ratio of the 70S to the subunit peaks in WT appears higher than in L5ΔPSL, especially in terms of peak area. This might not be ignored.

Authors’ answer:

As a matter of fact, in this paragraph we claim that the subunit assembly is not affected in the PSL mutant based on two observations: lack of additional peaks that would indicate accumulation of the LSU assembly intermediates and formation of the 70S ribosomes, which would be impossible if uL5 does not incorporate. This is further supported by the protein composition analysis. The amount of free ribosomal subunits may vary in the centrifugation experiment, but we also have a feeling that the subunits/70S ratio is slightly, but higher in the mutant. Very valuable remark, thank you. Although, such a moderate difference cannot contribute significantly to the gross phenotypic and functional defects observed, it’s certainly worth mentioning. We gently commented this in the text keeping in mind that appreciable subunit association defects look typically like ones published by Cukras and Green in 2005 (doi:10.1016/j.jmb.2005.03.075) and Bubunenko et al. in 2006 (doi: 10.1261/rna.2262106).

Text added to the end of the second paragraph of the section 2.2:

“At the same time, it is worth mentioning that the free subunits are slightly, but reproducibly overrepresented in the mutant as compared to the control.“

and to the fourth paragraph of the Discussion:

“We repeatedly observed that the free subunit content in lysates of the ΔPSL mutant is slightly higher as compared to the control strain. Although, such difference is negligible to be accounted for the observed physiological and functional defects in the mutant, it still may indicate that some small fraction of the ΔPSL LSU is somewhat compromised for the association with the SSU, at least, under the experimental conditions used to isolate and analyze ribosomes (low temperature, etc).”